# Vio-lense: A Model for Understanding How Violence and Senses Relate during Refugee Journeys in Europe, and How This in Turn Can Foster Collective Healing

**Georgina Lewis**

College of Social Sciences and International Studies, University of Exeter, Exeter EX4 4PY, UK;
ghl204@exeter.ac.uk

**Abstract:** Introduction: The 'senses'—our sight, sound, taste, smell, and touch—are intrinsic components of our human experience. Trauma recovery practitioners afford the senses to foster healing and draw into awareness the sensations of the traumatised body. Therefore, if the senses are valuable in trauma recovery, then they must also be important in the initial traumatic acts—such as violence. Aim: This ongoing PhD project explores the role of senses in violence committed towards refugees and asylum seekers in Europe. Method: A sensorial model is developed through semi-structured interviews and key engagement with literature and research. Results: An excess and absence of senses are critical in understanding violence endured by refugees and asylum seekers, but also in how violence is witnessed and perceived by others. It is clear people on the move experience violence sensorially throughout their journey. Conclusions: The senses are evidently part of the violence. At times, violence is in sensorial excess: the sounds of camps; the smell of tear gas. Alternatively, a concealment or deprivation of senses can also be violent: camps on the periphery; violence out of sight. Notably, senses (in excess or through concealment) can also be vital components in the survival of violence. "Vio-lense" is a suggested model for considering 'violence' and 'senses' as being essentially interwoven rather than separate. This, in turn, is important for development of collective healing mechanisms and, more widely, defining and understanding violence.

**Keywords:** refugees; asylum seekers; violence; senses; recovery; trauma; collective healing

## 1. Introduction

The senses—our abilities to see, hear, feel/touch, smell and taste—are an important tool for trauma recovery, with some trauma practitioners including sensory-based work in their practice and treatment models (May-Benson 2016, p. 3). Meanwhile, a review of occupational therapy literature detailed 18 papers providing evidence for sensory-based pursuits for patients who were survivors of trauma (McGreevy and Boland 2020). Senses can be used as 'grounding' techniques whereby patients are encouraged to utilise their senses to manage their anxiety and foster healing (Keeping Well NCL 2022; Smith 2018). Outside of formal therapy settings, engaging with particularly natural environments—such as walking through the forest—can invoke transformative healing encounters from the grounding aspect of the heightened sensorial experience (Murray 2017). Equally, using our senses (our visibility and audibility in particular) in performative roles, such as theatre, can establish spaces which foster collective social healing (Walker 2016). Performance is an intrinsically embodied experience which can help enable the body to symbolically and sensorially represent memories which may otherwise be too traumatic to recall (Walker 2016).

This paper builds upon this awareness of senses being important aids for healing, and argues for senses to be understood not only in the aftermath of trauma, but also in the initial act(s) of trauma. In particular, this paper is concerned with trauma resulting from violence, and aims to analyse if and how senses should be incorporated into understandings of violent experiences, as well as the aftermath. More specifically, this paper aims to

consider violence through a sensorial lens by examining the violence refugees and asylum seekers may experience on European journeys. The objectives of this paper are to outline preliminary results in ongoing research, and to analyse these results through a suggested model of 'Vio-lense.' It is hypothesised that throughout refugee journeys in Europe the violence, occurring at hands of official actors and policies, involves the senses, and that understanding this could improve chances of collective healing for people on the move.

### 1.1. Violence

An initial starting point for this research, and in continual consideration, is if/how one can define violence and in doing so understand what within refugee journey(s) in Europe can come under analysis. Mark Vorobej depicts just how complex and intricate conceptualising violence is, and how difficult—and not necessarily appropriate—it is to give violence a singular definition. Rather, violence exists as a concept of multiplicity. Vorobej explores various elements and forms of understanding and queries the traditional notion of violence being an *intent* to cause harm, rather than the fact of harm taking place irrespective of conscious intent. He questions the meaning of violence as a concept, and how this can shift depending on form of violence and social setting. Indeed, in a singular moment one may have concerns about violence in variously different ways depending on their social roles: I, for example, when out with my children, might have concerns about risks to myself as a woman, as a mother, for my children, and as a gay person. All these perceptions of possible violence are real, irrespective as to whether or not any acts of violence actually take place.

Karen Backstein calls for the various disciplinary approaches to defining violence to unite to try and manage this complexity (Backstein 1992). Various attempts for this are made, with Sean Byrne and Jessica Senehi's "The Violence Prism" offering a particularly compelling ecological framework through which to approach the multiplicities in defining violence (Byrne and Senehi 2012). Through their eight frames—psychology, socio-biology, structural theory, human needs theory, socialisation theory, feminist theory, anthropology, and international relations—they make a dynamic and considered review of the main theories defining violence but also of processes which may prevent violence. Overall, whilst the eight frames are by no means finite in defining and understanding violence, it strongly presents a case for considering violence as multifaceted and complex.

John Burton's portrayal of three "types of human motivation: needs, values and interests," is drawn upon as motivators for conflict—particularly when *needs* are not being met—which can escalate to violence (Burton 1997; Byrne and Senehi 2012, p. 37). From this, though, an approach to conflict, centred on the problem-solving of clashing or unmet needs, can aid in preventing escalation to violence (Burton 1997). The problem arises, of course, when policymakers consider having to 'hierarchise' the bodies who are all competing to have their conflicting needs met. The argument continues that if violence could be understood in such ways, then it might be possible for policymakers to work on more problem-solving approaches to manage competing needs and, thus, prevent violence (ibid.).

Whilst more traditional definitions of violence focus on intent, there is growing emphasis on the outcomes of actions as being important defining factors—for example the Galtungian theory of violence: that of violence being determined by harm (1969). Johan Galtung defines violence as follows: "When human beings are being influenced so that their actual somatic and mental realisations are below their potential realisations." (Galtung 1969, p. 168). In other words, if a person(s) is prevented from realising their physical or mental potential, or development, then this constitutes violence. For Galtung, action does not need to be direct, or indirect, to signify violence; it is the effect on potential development which is key.

Galtung's theory has come under some criticism. C.A.J. Coady criticised the broad nature of the definition, but Galtung acknowledges this as being a necessary aspect of broadly encapsulating violence (Coady 2008; Vorobej 2016). Coady also alludes to some everyday examples of life to query the legitimacy of Galtung's definition—for example that

of a child causing a parent to become tired, and as such preventing their physical potential. Vorobej defends Galtung's theory of violence as being based on overall wellness and development, not one component such as fatigue, and of examples whereby the restriction on development and realisation was not outweighed by the growth (Vorobej 2016, p. 87). For example, in the case of the child causing parent tiredness, one can probably assume the benefits for the parent of having their child far outweigh the impact of fatigue. As such although this one aspect of their potential realisation has been hindered, overall, their development as a human being has flourished in their life as a parent.

Continuing to explore violence through a Galtungian lens, we also need to consider the difference between personal violence and structural violence. Personal violence is when there is a recognisable human, a visible actor, responsible for restricting potential realisation (Vorobej 2016). There is usually a clearly identifiable perpetrator, and victim, and more often with intent to cause harm. Structural violence, meanwhile, continues with the definition of restricted potential realisation, but without the clear responsible actor, or necessarily the intent (Vorobej 2016). An example being if a group within a society were unable to access the same resources necessary for optimal survival and development as another group (Vorobej 2016), then this would constitute structural violence. It is not necessary for there to be intent for there to be structural violence (Vorobej 2016). On the contrary, it is rarely a direct outcome of premeditated harm. The difficulty here is that it becomes almost too easy for structural violence, existing as it does along a continuum, to become perceived as a necessary evil for the greater good in a society (Vorobej 2016).

An extension of the Galtungian theory can be found in Vittoria Bufacchi's definition of violence, as a *violation of integrity.* Their overall definition of violence is as follows:

"An act of violence occurs when the integrity or unity of a subject (person or animal) or object (property) is being intentionally or unintentionally violated, **as a result of an action or an omission.** The violation may occur at the physical or psychological level, through physical or psychological means. A violation of integrity will usually result in the subject being harmed or injured, or the object being destroyed or damaged." *Emphasis added by myself.* (Bufacchi 2007, p. 43/4; cited in Emerick 2019, p. 31)

I have added emphasis here because when researching violence, and regarding the senses' relationship to violence, that an action *or* an omission can both be violent is essential in definition. Senses can be an action: making sound; forcing a scent; or witnessing an event, but senses can also exist in omission: loss of sound; having sight forcibly removed; not being heard.

Barrett Emerick's theoretical development of violence as breaking beyond the "traditional" binary between physical and psychological violence into the realm of epistemic violence is particularly compelling (Emerick 2019, p. 33). In this definition, Emerick is focusing on a person's capacity to *know*—to have knowledge and to establish meaning from knowledge—as being something which can be violated to such an extent that a person suffers epistemic violence (Emerick 2019, p. 33). Like Galtung, Emerick focuses on an outcomes-based approach to violence—if an act (or omission) results in a violation of a person's knowledge and integrity, then that is violent regardless of prior intent. Emerick includes acts of violence commonly referred to when considering violation—sexual offences, being threatened, violating self-worth and dignity. However, what is particularly relevant to this paper is how Emerick furthers this by expanding on the notion that *silencing* a person(s) can also be a violation of "a person's epistemic capacities," and can, therefore, be acts of violence (Emerick 2019, p. 33).

Thus, the gap between senses and violence, rather than being considered just in the aftermath as a mental health support, exists in the theory of violence itself. Emerick examines how silencing can violently occur both at an interpersonal and institutional level, and within that both in overt and covert frames (Emerick 2019). Further, if being silenced is continual over time, not only may persons experience the violence of marginalisation and Othering, but they also lose their own sense of self-worth with their own knowledge—epistemic violence *through* silencing. When considering the grounding mechanisms in

trauma recovery, practitioners must therefore be aware of any overt or covert silencing which may have happened. This is difficult to uncover, both as a practitioner and researcher, because silence by its very nature is an absence—a deprivation of sound. Trying to bring absences back into tangible knowledge is a complex task but, nonetheless, has a growing body of researchers attempting to draw out theories and understandings of how we may research what we cannot see, hear, or know (Rappert 2015; Paglen 2009).

I am examining this relationship between senses and violence for refugees and asylum seekers in Europe and the journeys they undertake. The violence that is under investigation is that committed by official actors—police, coastguards, doctors, and anyone else working in an official capacity. Whilst this is an ongoing piece of research, this paper seeks to expand on some of the preliminary findings. A model of 'Vio-lense' is established by furthering theories and patterns found in participant interview transcripts; the paper concludes by establishing how understanding the sensorial experience of those in Europe can aid in strengthening therapeutic healthcare and healing support. Firstly, we consider some of the theories and arguments involved in senses and violence, before outlining the methodology. A results section follows, before a more in-depth discussion ending with how understanding the relationship between senses and violence can aid collective healing.

*1.2. Senses and Violence*

Whilst we exist in a world of multiple meanings embodied in many cultures, the human body's capacity for senses is an intrinsic aspect of our existence. Whilst Western societies do generally adopt an emphasis on our sense of vision, it nonetheless remains that the human body is cross-culturally capable of perceiving and emitting various senses and creating epistemic knowledge and meaning by doing so (Le Breton 2020). One way our senses are experienced differently is in how we can communicate—through our language— and importantly how one person sensorially perceives a situation may differ from someone else right next to them (Le Breton 2020). Nowhere am I aware of this than when travelling in our car—whilst I am visually paying attention to the immediate settings around me for any risks, my children are busy spotting (and shouting out in narration at random haphazard moments) visually 'exciting' pieces of the mundane that I, apparently, simply must know. So thus, whilst I see traffic lights changing to red, my son has hollered "wow a lorry!" whilst my daughter has, for the tenth time in the last minute, informed me that there is a tree. This tongue-in-cheek analogy is to example how in one same second with the same visual information before us, three different people perceive three very different realities (and have different perceptions on who needs an auditory narration)! As David Le Breton aptly describes, "[Vision] only ever apprehends one version of an event." (Le Breton 2020, p. 47).

Oftentimes, our bodies can be experiencing a different set of sensorial information than we psychologically may be processing. This can be particularly true with people who have experienced trauma and may suffer with reliving symptoms such as flashbacks and nightmares. In these events a person may feel that they are physically experiencing that trauma again in the here-and-now, because the sensorial experiences in the body are so terribly vivid (PTSD UK 2023; CPTSD Foundation 2019). This is, again, why in therapeutic settings the use of senses to 'ground' patients is so essential for their continued health and wellbeing. My experience as a patient within the NHS, and previously a mental health worker in an NHS psychiatric hospital, has shown me there is a very intrinsic need for senses to be understood and utilised in trauma recovery (Lewis 2017). Often, 'grounding mechanisms'—tools to 'ground' us into the present moment—utilise our bodies and our perception of senses to restore our bodies and minds to the safe present and away from distressing memories and feelings (Keeping Well NCL 2022; Smith 2018). Leaning against a wall; shifting weight on our legs; squeezing ice cubes in our hands; counting a certain number of coloured objects; listing what we can hear or smell—these are all such examples' professionals may encourage patients to use as grounding mechanisms, and all of which afford the senses (Smith 2018; May-Benson 2016). Certain therapies may focus on some

senses more than others but nonetheless for many people healing from trauma requires engaging with our senses (McGreevy and Boland 2020; ibid.). In many cases, such trauma can and does include memories of violence.

Senses can also become unpleasant. Particularly is this the case with our sense of hearing and our olfactory senses. It is difficult (although not impossible) to escape and excess of sound—so too is it difficult to 'switch off' a potent smell. Whilst these moments of repulsive experience are not often associated with violence, this paper seeks to bridge this gap between senses and violence. Indeed, we can consider briefly how racism can and does use the senses to 'justify' violence—the sense of sight to ostracise those with different skin colours, or to accuse the Other as having a strong smell in a way which dehumanises to the extent as comparing to animals (Jones 2021). An example of this is described by Hannah Jones regarding a news outlet in Kos, Greece in 2015. They reported the "sight (and sounds and smells) of border violence"—too sensorially overwhelming for holidaymakers to ignore—as "disgusting." (Jones 2021, p. 61) The sensorial excess for the tourists was "too much and too disturbing." (Jones 2021, p. 61). As a strategy for their own selves, and to preserve the idyllic ignorance which tourists seek on a holiday, people chose to ignore and look away from the Other. Indeed, in response to the sight of a camp, restaurants built a net barrier "to block the sight." (Jones 2021, p. 62).

Hannah Arendt builds a more theoretical understanding of visibility regarding the humanness of refugees (Arendt 1951). For Arendt visibility is a social practice, inasmuch that the social political world is one of appearances (Brambilla and Pötzsch 2019). Arendt argued that the 'stateless refugee' has lost their position within a visible social appearance—they are no longer citizens, able to perform their political action (Borren 2008). By losing their citizenship, and therefore their political status and right to *public visibility* then their status as a human alters (Arendt 1951; Borren 2008). Arendt's argument is that they are not reduced to subhuman but reduced to being visible only as a 'natural man'—human in its most bare, natural form, back to homo sapiens (Arendt 1951; Borren 2008). They are "nothing-but-human ( . . . ) human-all-too-human," (Borren 2008, p. 219). This removes their capacity to be publicly *visible* which is what Arendt argues as key to political action and participation (Arendt 1951); where for the citizen with rights to live a political enacted life means being publicly visible and naturally invisible (protected from state scrutiny and intrusion), for the refugee this has been reversed. Their shift in human status leaves them publicly invisible—branded as a one-group, nameless, a bare-naked human body—and naturally visible—not protected from intrusion or with any right to privacy (Arendt 1951; Borren 2008). It is the instinct to assume a dehumanisation requires a *loss* of humanness. This theory challenges that notion. Rather, a dehumanisation can be the process of becoming too *much* human. The violence of that dehumanisation is intrinsically connected to the sensorial visibility of a person(s). Arendt's key work on this was written in the post-WWII era. It is nonetheless still highly relevant today (Borren 2008; Köhn 2016) with "disturbing parallels" between the action towards refugees in Europe then, and now (Borren 2008, p. 225).

*1.3. The So-Called 'European Refugee Crisis'*

In 2015/6, I travelled to the 'Jungle' in Calais. I was immediately overwhelmed by sensorial excess. As I stepped into the camp, I no longer recognised 'France'. It had been raining unforgivingly for many days now, and the ground was thick with mud. Toilets—the responsibility of authorities—had been neglected and were overflowing onto the ground. Everywhere there was noise—the whirring of generators, the chattering of multiple languages, the call for prayers, the singing. The smells ranged in mere moments between burning plastic, faeces, to the delicious smells of traditional foods being cooked in makeshift cafes and restaurants. Visually everywhere I looked there were tents soaked in mud. Even in this I saw different perceptions of meaning—whilst adults walked around holding themselves carefully to reduce as much mud splashes as possible, I saw a small child happily jumping around in oversized wellies, a huge smile on his face. In hell, but to

this little one, a fun playground. I had to take a few moments to ground myself, using my own senses, as I felt a rising anxiety at the conflict of vastly ranging sensorial messages. Later I would hear how the smell of tear gas leads to children coughing in the night, and I would witness for myself how the intimidating presence of the CRS Police[1] would lead to sudden silence in the immediate area.

Giorgio Agamben considers the 'camp' to be the space containing those living in bare life form. The inhabitants are subject to a new kind of prison guard in which the rules are ever-changing, and they are not protected by any legal status (Agamben 1998). If we consider this further under Foucault's notion of heterotopia, of camps being out-places on the periphery of society (Foucault 1978; Agier 2011), then the EU have pushed the borders of heterotopia quite literally out into the water waiting for someone to claim responsibility for those stranded upon it. And this shifts, almost continually; each day I open my computer to read more testimonies from NGOs across Europe, posting on their social media the latest atrocity that we are unlikely to read about in mainstream news. The status and geography of 'camp' for refugees is fluid; it changes fast, is both borderless and yet bordered. The camp as bare life, and heterotopia, is constantly shifting its own border; the refugees are imprisoned outside but the dimensions of that 'outside' changes rapidly (Foucault 1978). What remains consistent is that senses continue drawing meaning irrespective of borders, and that a sensorial perspective of the situation in Europe captures acts of, and implications of, violence.

### 1.4. Literature Review

Despite mainstream media by and large failing to adequately report on the violence(s) inflicted throughout Europe, there nonetheless remains an incredible level of documentation made by committed activists, NGO workers, volunteers, academics, lawyers and more (Forensic Architecture 2020; Human Rights Watch 2017; Petkova 2016). Through huge dedication and commitment, violence has been documented in written form as well as through photographic and film medium, often to the risk of their own safety (WatchTheMed 2021; Barker and Zajović 2022). In many ways, the violence is committed in direct and deliberate approaches, but often also the violence is an indirect outcome or result of willful ignorance or policy. Hannah Jones (2021) establishes a compelling, and (necessarily) challenging, concept of violent ignorance—whereby this purposeful choice of ignorance (and I would further, of choosing a sensorial absence of 'looking' the other way), even when for self-preservation, can lead to further violence. Further, that acts of ignorance can be committed violently. One example is the Grenfell Tower disaster, wherein residents' complaints and concerns about the structure of the building had been ignored repeatedly by the authorities, and ultimately this ignorance led to the tragic loss of 72 people. Jones described this scenario as a "clear case of institutional indifference and violent ignorance," (Jones 2021, p. 55). In this chapter, and indeed throughout her book, Jones entwines issues of the senses within the understanding of ignorance and its relationship with violence. Regarding the Grenfell Tower disaster, Jones includes a section dedicated to unpicking the notion of "silence" (Jones 2021, p. 57). The Grenfell Silent Walk—taking place on the 14th of every month—is analysed for how the use of "sustained and dignified" silence "in the midst of noisy London streets" makes it a silence that "cannot be easily ignored." (Jones 2021, p. 57). Sometimes, as with myself all those years ago, silence is an outcome of violence which paralyses the oppressed. Other times, as in this situation, silence is an outcome of violence which powerfully alludes to the unspeakable and has empowering qualities. The silence and ignorance of the authorities resulted in horror; the silence during the walk is an *active* silence, the antithesis of indifference.

One immense project documenting the violence and abuses in Europe is the "Black Book of Pushbacks" which totals over 3000 pages of testimonies concerning state violence across Europe towards refugees and asylum seekers. It particularly focuses on violence at the hands of 'official' actors such as police, coastguards, and border patrol forces. Violence is detailed, with ample evidence, in many countries (Barker and Zajović 2022). A particular

issue was the impact of COVID-19, resulting in many NGO organisations needing to leave the field to protect themselves from the risk of transmission. This had far-reaching consequences for people on the move, with many countries across Europe opting for "deterrence tactics" involving a great scope of physical and mental violence. Some examples are forced undressings in freezing temperatures, sexual assaults and beatings (Barker and Zajović 2022, p. 5). We can appreciate here the sensorial acts of violence and consequences of that violence—for example the forced undressings not only violating dignity and psychological welfare but also, oftentimes in freezing conditions, being physically painful and at times even fatal—for example on one occasion 19 people freezing to death as a direct result of forced undressing (Barker and Zajović 2022, p. 5) The use of 'zip ties' before being thrown into rivers, to physically prevent people from being able to swim and/or reach safety, is also documented (Barker and Zajović 2022). This violence begins with sensorial excess (the sharp pain of wrists being ties) but establishes sensorial absence (preventing person(s) from being able to move or use their arms). This is an example of how multiple forms of violence relates to senses in various ways—representative of the complex and often multifaced definitions of violence.

"The New Internationalists" (Clayton 2020) is another compelling and vital work containing many first-hand testimonies from volunteers, workers, and other witnesses to the violence in Europe. Brendan Woodhouse, a UK volunteer in Turkey, described the scenes of a "rubber boat" hitting rocks. The description is rich in sensorial information—for example the auditory awareness of screams, the visual element of phone lights, followed by darkness as the phones fell in the water and silence as people went overboard (Clayton 2020). Woodhouse then describes entering the water himself on a rescue mission: "The first bag I came to, that was it—a small baby. Her eyes were in the back of her head, her lips were blue." (Clayton 2020, p. 71)

Woodhouse's testimony illustrates how people's memories of violent and devastating events such as these vastly involve the senses. In one memory there is the conflict of sensorial excess (screams and yells) with sensorial absence (silence and darkness). There is meaning in the description of senses; tragically we know what "her lips were blue" means, and even if Brendan is unable to say the words directly, his (sensorial) description provides us, the reader, with the knowledge that the baby is not breathing. Another worker, Barbara Held, recalled having limited visual knowledge due to the dark, but understanding from the sounds that people were panicking. This displays how the loss or inhibition of one sense means that we use other senses to acquire knowledge and meaning even in the most difficult of circumstances (Correct—Clayton 2020, p. 113).

Information about the violences in Europe is also readily documented on various social media outlets in particular trusted and registered NGOs on the ground. These organisations are able to share information quickly about events taking place and also ensuring that those lost or 'missing' in Europe are made known about. For example, Aegean Boat Report—an organisation which works to document the ongoing situation in the Aegean Sea—described in April 2022 the scenes of a child's body being discovered in the sea—too much in decay to form an identity (or even sex) but at least able to ensure that whoever that child was, they were acknowledged (Aegean Boat Report 2022). That this child had been missing for enough time to decay in European water to an unrecognisable extent establishes how the child is visible and yet invisible in identity. Reece Jones (2017) explains how there is an "invisibility" to the bodies "lost at sea" (Jones 2017, p. 18). However, the bodies of those who drowned have not simply ceased to exist. Not washing ashore and becoming tangibly visible does not also insist that they are *wholly* invisible; the bodies—in different form—remain in the sea, in the floor of the sea, and in everything the sea touches. Being at sea, with only some organisations such as Aegean Boat Report able to bring to awareness the reality, make it easier for individuals and communities to remain violently ignorant. "It is unprecedented. And yet it is already normal." (Berger and Mohr 1975—cited in Hannah Jones 2021, p. 69).

Overall the literature supports the notion that the violence(s) in Europe involve senses in multiple and complex ways in which an excess of senses can also entwine within sensorial absence. It is the aim of this research project to more comprehensively draw these concepts together in theoretical analysis.

## 2. Materials and Methods

### 2.1. Type of Study and Study Design

This is a qualitative study with interview technique study design. I conducted semi-structured interviews with participants without seeking to establish scientific causation. Rather, this research offers a view of the situation on the ground in Europe and an analysis of what that could mean for understanding a relationship between violence and senses. Using a phenomenological approach during the interviews allowed for participants to discuss how they interpret the relationships between senses and any violence that they are aware of. Semi-structured interviews were necessary to ensure a similar 'guide' across the study but also to allow for individual differences depending on the role of the participant in the field, and the geographical location they were discussing.

Interview transcripts were coded using grounded theory approach, and analysed inline with the research hypotheses.[2]

### 2.2. Setting

Semi-structured interviews took place with participants over digital formats such as Zoom or Microsoft Teams. All interviews took place from the UK but participants were based in different locations (see Table 1). Furthermore, most participants had volunteered/worked in at least 2 different settings so were able to expand on their knowledge of different areas. Recruitment lasted from mid-2021 until early 2023. Participants were involved for one interview session. Interviews took on average an hour. Participants were provided with a transcript of their interview with a three month period in which to respond.

**Table 1.** Initial participant data up-to January 2023.

| Location Discussed/Experienced | Participants | Sex | | |
|---|---|---|---|---|
| **GB** | **17** | **Female** | **Male** | **Total** |
| Greece/islands | 6 | 23 | 7 | 30 |
| France | 5 | | | |
| Italy/islands | 4 | | | |
| Germany | 4 | | | |
| Belgium | 2 | | | |
| Sweden | 2 | | | |
| Turkey | 2 | | | |
| Hungary | 2 | | | |
| Ukraine | 1 | | | |
| Netherlands | 1 | | | |
| Austria | 1 | | | |

### 2.3. Participants

i.  Eligibility criteria

Interview participants were sought who fit the following criteria: having supported and/or worked with refugees/asylum seekers in Europe since 2013; with ability to read and speak English well; with access to technology such as Zoom or Microsoft Teams.

ii.  Exclusion criteria

Participants were excluded if they were not legal adults (aged 18+). Participants were also excluded if they could not provide sufficient consent and clear understanding of the research project. Refugees/asylum seekers and other people on the move were excluded from participating due to ethical concerns. Participants needing a translator were excluded

after considering the limitations of technology, in particular potential issues with several people in one conversation (connectivity, etc.) if a translator was included. However, there were also ethical concerns of multiple people being present in digital interviews and it being therefore harder to ensure confidentiality and safety.

iii.    Participant data

Table 1 establishes some demographic information about the participants. Certain data such as 'age' are omitted for the purposes of participant confidentiality. Sex was not specifically asked by researcher so it is important to note that this data included is assumptive. Whilst 30 participants are included, their experiences illustrate 47 different settings. However, it is important to emphasise that this is not of statistical significance as participants did not all discuss settings with precisely the same depth of information and breadth of knowledge. Over half of participants were able to refer to experiences in the UK, owing—at least in part—to that participant requirement of speaking English fluently affecting eligible participants.

iv.    Recruitment

Participants were recruited online. Relevant organisations were contacted directly with an opening informative email and attached information sheet detailing the research aims. Organisations were invited to circulate the email amongst their colleagues if felt appropriate. Recruitment began in the UK before establishing connections with organisations in European countries. Participants were invited to provide details of the project to anyone they felt may be interested. Some organisations responded explaining that the pressures on the ground were too great to consider research, and/or that they received too many research requests to manage. However, on the whole, response was positive with a keenness to participate in work which may aid further in sharing awareness of the situation.

v.    Study size

Thirty participants have taken part to date with research ambitions to reach 40 participants. Whilst this number may seem low when attempting to reflect the picture across Europe, the quality and breadth of the participant knowledge has been greater than anticipated. Most participants had volunteered/worked in two different settings and were able to provide data for each of these. In some cases, participants were able to discuss up to four other settings.

*2.4. Methodology*

The methodology for this research has shifted following changes to research practice due to COVID-19 restrictions. Initial plans were to volunteer in the field, and subsequently hold face-to-face interviews with refugees and asylum seekers at various points in Europe. Following the onset of the COVID-19 pandemic, this became highly unlikely for the foreseeable future, so in the interests of time restraints and a need to begin research, the methodology was adapted.

Participants took part in an online semi-structured interview pertaining to their roles (previously and/or currently), their experiences or understanding of violence in Europe (towards refugees and asylum seekers, although at times this violence was directed at frontline staff in the field too) and then drawn into considering the implications of violence with the senses. Participants were encouraged to consider their own meaning and perception of senses and violence in these settings. Interview transcripts were coded using NVivo following grounded theory approach (Urquhart 2012; Gibson and Hartman 2013). I approached each transcript, and any literature containing first-hand experiences, following the methodology suggested by Murray Schafer (1992)—to consider each document in turn and make note of any references made to the senses. In Schafer's case this was specifically regarding sound, but I expanded on this to seek out any references made to the other senses as well. This examined clear references to senses, but also implication of senses—for

example, of people being 'made missing', or the discussion of refugee voices being made more audible.

### 2.5. Ethical Considerations

All participants were provided with consent forms and information sheets prior to participation. Participants were made aware of their right to withdraw at all times. Confidentiality was a high concern of many participants. Each participant has been provided with a pseudonym and names of organisations are omitted—where needed the general purpose of the organisation is included to provide context. People on the move were not included due to ethical concerns regarding their safety via online interviews, but also the risk of retraumatisation following the discussions with researcher at too great a distance to safely manage.

## 3. Results

As this is a qualitative research project, the results here are not to experimental or of statistical value. It should also be re-emphasised that this at present is a preliminary findings paper of ongoing research. Findings thus far are sufficient to draw some compelling conclusions and initiate the reasoning for the 'Vio-lense' model (Table 2).

**Table 2.** 'Vio-lense'.

| Sensorial Excess | | Sensorial Deprivation | |
|---|---|---|---|
| **Overt** | **Covert** | **Overt** | **Covert** |
| Example: Use of tear gas | Example: Overflowing toilets | Example: Destroying water containers | Example: Children missing following camp destruction |

### 3.1. Vio-lense (Table 2)

From the interviews, it was apparent that there are various ways in which senses are implicated in the violence across Europe. The following table was developed to best approach analysing the data. This table was established after reviewing the data and becoming aware of growing patterns, and this is the basis underpinning 'Vio-lense' (Table 2).

By 'sensorial excess' this is regarding situations where one's senses are overwhelmed to such an extent it could be considered violence. This can happen in overt, direct, ways—such as police use of tear gas, or beatings at borders—and it can also happen in indirect ways such as the absence of moral actions. Examples here could be toilets left to overflow in refugee camps, whilst which sensorially excessive, is the outcome of a deprivation of action. Another example would be camps being placed in locations where the worst of weather fronts hit, without provision for ample protection.

In a similar vein, 'sensorial deprivation' is regarding situations where one's senses are deprived or made absent to such an extent it could be considered violence. Again, in this sensorial deprivation, it is possible for this to be overtly violent in causing absences—such as police destroying water containers, and thus refugees' capacity to access the taste of clean water, or covertly violent in its absences—such as police neglecting to safeguard unaccompanied children (resulting in many missing) after a camp is 'cleared'. The sensorial absence here is the loss of visible, audible children with no awareness as to their location, safety, or indeed assurance of life and the all-to-real knowledge that their visible, audible selves should in fact be present. All the above examples were either witnessed during my fieldwork in Calais (Lewis 2016) or reported in present interviews. 'Vio-lense' is drawing upon the notion that violence involves violation(s) and that a lens in which to consider such violations is through that of our senses: vio-lense.

### 3.2. Location

How this table appeared varied depending on geographical location discussed. A general pattern emerged whereby entry points to Europe—the Aegean Sea and Mediter-

ranean entry points, for example—largely emphasised violence in sensorial excess, whereas once through Europe and particularly into the UK, the violence shifted more so towards sensorial deprivation or absence. Frequently sensorial descriptions included one or more aspects of the above table in a single episode.

(A)   Entering Europe—The Mediterranean

For many, the journey across the sea remained by far the most traumatic—for those on the boats, and for the witnesses. Interviewees and testimonies depicted the sensorial contradictions with the silence of the drowned, and the cries of the survivors. Whilst the silence, and the cries, are in themselves not violent—they are born out of violent European policies resulting in loss of life and trauma. Participants also discussed the issues of illegal pushbacks—whereby people arriving into Europe are physically pushed back into the water by apparent official actors, often in directly violent approaches and with catastrophic consequences.

Participants acknowledged how people on the move often struggled to hold conversations about these particular traumas. For many, it was impossible to put into words. Participant SF details, "He did the boat trip ( . . . ) and he said that was one of the worst experiences he's ever had. He still dreams of it." For many survivors, talking about what happened—particularly in the boats—is not possible, but the absence of that narrative serves to illuminate the known presence of trauma. Many choose not to talk; many others are told now is not the "right time" to talk depending on where they are in their asylum journey. This can lead to a situation where nobody is considering themselves qualified to handle such conversations, nobody asks, and the "right time" does not appear:

> "The general consensus is now's not the time for them to process that trauma ( . . . ) now's not the time, now's not the time. That's something that really struck me. When is the time? Like, at what point is there space for mothers to grieve losing their children at sea? Where and when is that time given to them?" *Participant MS.*

This was something which became apparent in several conversations. When asked if people had disclosed much about their journeys the answer was either not, because it was too traumatic, or because it was not the "right time"—although who was deciding when this right time was remained unclear. This may not immediately seem sensorially relevant but in fact the inability to audibly verbalise one's experiences because they are too horrific to manage in words is a significant aspect of the violence. Sometimes, when asked or when people on the move tried to offer information about their journeys (particularly again regarding the entry to Europe) they elicited a clear trauma response:

> "The reaction I got was he was holding himself so tense, particularly when it came to certain things on that journey. He only said two or three words. But I could see he was very close to tears, breaking down." *Participant DC.*

This shows how traumatic memory can be sensorial not just in its account, but in the act of physically remembering—the tension of the body, the closeness of tears.

Participants also described the loss of life at sea. Participant JB described:

> "They came 40 people in a boat, which is supposed to be for 4 or 5 ( . . . ) A pregnant woman gave birth in the middle of this, and it was a dark night. They couldn't keep the child quiet. And there were some coastal guards around, so they took the child and dropped them in the water."

The traumatic violence here is multi-layered with sensory implications—firstly that a woman had to give birth in such grave conditions, presumably in great pain and fear. Secondly, that they were in a dangerously overcrowded boat, forced to travel by night to try and escape the surveillance gaze of the coastguards (for fear of pushbacks or worse). And then most tragically, due to the natural sounds of her new-born, a life lost out of fear that the coastguards would hear the baby and thus find them. This is an example whereby sensorial survival came at huge cost. The need to remain invisible and inaudible—to try

and make it safely to land—meant that in order the 40 people aboard survived, a decision was made to silence the baby. Participant MS similarly described a mother losing her two-year-old who fell off the boat into the water and stopping was not possible. Other stories were heard of children being mistaken—in the dark chaos of a sinking boat—as 'bags' and thrown overboard.

Participant MS described how a person she'd helped described being on the boats:

"All she could hear was howling because everybody was scared, they were grieving ( . . . ) the woman who lost her children was howling. Everyone was howling ( . . . ) when they were on land, everything was quiet again, but she could still hear the howling ( . . . ) the screaming didn't stop for her."

In this example, we are made aware of the sensorial excess—the screaming—which is a result of covert violence (policy; SOS calls left ignored), followed by the sensorial absence and loss of sound. Except, for the woman in question, the traumatic memory meant she continued to hear the screaming for some time. This is one example of how violence can impact and affect the senses far beyond the actual event.

(B)    European refugee camps

A considerable number of the participants had volunteered/worked in some capacity in at least one refugee camp. Most notably discussed was the Calais Jungle and the aftermath of its destruction, and the situation in Dunkirk.

Participant RW described a recent (at the time[3]) destruction of the Dunkirk camp:

"There was no warning, at all. They had no idea what was going to happen. And overnight it was raised. Literally the whole place. All the tents were taken. ( . . . ) They didn't just move them on. They took all of their belongings, which is not very much no, but essentials . . . they took them, and blankets, anything that was left behind, and thrown into the rubbish. It wasn't given away. It wasn't given back. It was just thrown away. So the police went in followed by these rubbish trucks. Right in front of everybody who lived there."

Here, we see clear examples of violent sensorial deprivation leading to sensorial excess– the destruction of sense of touch (warmth, safety) leading to an excess of cold temperatures. There is meaning to be made, too, of the violent symbolism of police coming in followed immediately by rubbish trucks—of how dehumanising the intention of this sensorial destruction is. RW explains further the implications:

"A lot of them don't have mobile phones. We can't contact them. So it's still emerging where they're gone. They [police] didn't say anything about it. They haven't released a statement about it. They haven't told anyone what's happening. They just raised the camp."

This sensorial absence has been exemplified—people who had been living in the camp are now missing and unaccounted for, and the perpetrators of the destruction are silent. Finally, RW criticised the language used to describe their actions:

"It's the way it's phrased . . . the police cleared a camp, it sounds quite gentle. It's the opposite of gentle. It's not the camp just being cleared, making tidy or whatever. It's like the camp didn't exist. It's just wiping away its existence."

And here, we see the final implications of the manipulation of language to depict sensorial 'gentleness'. The 'clearing' of the camp appears as a violent act to sensorially wipe evidence of it having existed at all. That such language—clearing—was used suggests that on some level, those ordering the camp's destruction are aware of the violence and making attempts to lessen that knowledge.

During the COVID-19 pandemic, authorities implemented strict rules for many refugee camps whereby the volunteers and workers were no longer permitted access. This was not only in France but elsewhere such as Greece. Participant ES voiced concern not only of COVID-19 outbreaks in such crowded and often squalid conditions as refugee camps, but that—in a similar vein to above—any violence now had minimal 'official' witnesses.



Indeed, several of the participants believed that the pandemic had become a convenient 'excuse' for authorities to force people on the move—already living on the periphery of society—to be unreachable. Participants shared concerns that access had been extremely difficult, and that ultimately the lives of Europeans needing kept safe from COVID-19 came at the expense of the lives of those trapped in overcrowded refugee camps. These policies, whilst not directly violent, nonetheless increased the invisibility of those in camps and of what may have been happening to them.

(C)   UK

In discussions about the UK, there was a sharp decline in evidence of overtly excessive sensorial violence (although some still existed, particularly in detention centres), but a staggering awareness of refugees and asylum seekers who under governmental policy, seem to disappear across the country: "Well, when I say disappearing, they *got* you know … disappeared." *Participant DS.*

By this, almost every participant who had worked or volunteered in the UK knew of a refugee or asylum seeker who had been kept 'on the move' and not given opportunity to settle. Often, they were moved under the guise of 'safeguarding', but with confidentiality policies meaning that the organisations who had been supporting said person were not made aware until, and unless, the moved person was able to communicate to them. As not all of them had a phone, this was not always possible. A particularly worrying occasion was reported whereby a 39-week pregnant woman, who had been receiving maternal care through the NHS, was moved several hours away without referral to new local maternal care. 'Luckily' this woman was able to contact her previous midwife, and remotely receive support—but not before going into labour whilst unaware of where she was or what she should do.

Another participant described the frustrating situation whereby a man he'd been helping had been registered onto a carpentry course in Wales:

"And he had done extremely well with that. And he was a week away from taking his final exams, when the Home Office decided, he was going to go and live in Croydon. He was literally picked up one morning from Cardiff and moved, because that was the time, they had chosen he would move. There was no consideration about the fact that actually if we left one more week, he'd have a skill. A bit of paper he could use in life. That was gone."

Whether this decision had been made deliberately, or as an oversight, was unclear but the implications remained the same. After spending two years working on this qualification, the man was again somewhere completely new, away from support, and with his hard work lost. This violation of his integrity and self-worth could well be argued as an act of violence, through sensorial absence—the absence being his loss of present voice, and his visibility at being moved. That the UK policy seems to regularly move people and return them to a state of invisibility has far-reaching psychological and physical consequences.

Meanwhile, in the midst of the COVID-19 pandemic the UK Government opened 'Napier Barracks'—a disused army barracks used for housing refugees and asylum seekers. The conditions were dire. Participants expressed concerns regarding safety:

"There was just no Covid safety ( … ) they knew they had Covid, and there's nothing they could do … they couldn't keep themselves safe, because all that was separating each other were these sheets." *Participant FS regarding the crowded state of living inside Napier Barracks.*

At times, it was reported that individuals chose to sleep outside, by the barbed wire, to try and protect themselves from COVID-19. Despite the pandemic, asylum seekers were placed in over-crowded dormitories with 28 people being expected to share two showers and two toilets. By 22 January 2021, over a quarter of the population had COVID-19, with 22 individuals on suicide watch.

Here, people are living in a world of sensorial excess—forced into crowded conditions, with minimal protection from a potentially fatal illness. However, this—as with most of the

reported violence and violent policies in the UK—happens in a covert manner; it is policy, rather than physical hands of guards, enforcing such crowded and dangerous conditions. This issue of vio-lense and health is prevalent across Europe.

*3.3. Sensorial Protection*

Whilst the above results have illuminated how vio-lense can work as an analytical model, another more positive finding was clear in the interviews. Whilst senses can be used violently and/or have violent implications, refugees and asylum seekers are of course not passive victims either. Many of them are highly resourceful and adept at surviving awful situations, and this extends into their recovery and aftermath also. There were several examples of people utilising their own senses—in excess and together also in absence—to protect themselves on their journeys or once in the UK. For example, many participants discussed how people on their journeys often existed in a nocturnal pattern, using the visual security of darkness to travel whilst sleeping during the day to avoid the gaze of authorities.

Another, quite touching example, was described by Participant DC. This was pertaining to young men in the UK who had set up their own football team, which for many reasons was a healing as well as fun activity to be part of. However,

" ... And they come [to practice] on the train. And what they were finding is they were being—I'm not sure if railway police or actual police—but they were being questioned. They [police] were questioning the trip. They came up with a very good solution, because they asked the charity ... could they have football shirts with numbers on their logo? So then, if they were all in the same uniform, they will believe that they are going to a football match, then the police would leave them alone. And they did!"

This was a fascinating example of using senses—in this case visual senses—to provide safety and security through a uniform. With a uniform their presence on the train came with approved of social meaning and whatever stereotypes the police held previously were kept at bay. The young men understood that by utilising senses in this way—of increasing their visibility in a socially acceptable manner—they would be protected. Whilst such efforts were needed in the first place is a separate issue, it was stories like these which gave a different insight into the sensorial world surrounding violence for refugees and asylum seekers.

## 4. Discussion

*4.1. Vio-lense (Table 2)*

Vio-lense is a model established in this research for analysing accounts relating to violence and the senses. The outcomes-based approach to exploring violence in this way supported and expanded upon previous research by Barrett Emerick regarding covert and overt violence and violent silenc(ing) (Emerick 2019, p. 30). For the purposes of this research, violence under analysis included any enacted by, or consequential to, official actors and policies.

Throughout the interview and analysis process, I was keenly aware that for every image, sound, smell described, there were so many more not borne witness too—or, indeed, witnessed yet too awful to be recounted. These absences just speak into being the presence of voices, memories and stories untold, which is a nature of narrative. It would therefore be insufficient to claim the conclusions drawn (so far) in the research can be applied generally throughout and across the entire so-called 'refugee crisis' in Europe. The findings nonetheless contribute meaningfully not only to the fields and knowledgebase of refugee studies, violence studies and sensory research, but also more broadly to that of collective healing.

### 4.2. Absences

Vio-lense is a useful model to analyse not just the described violence, but also the absence of description. People can refrain from talking as a method of self-preservation, but their bodies can display sensorial implications of their memories. Sometimes absences are a vital part of the memory—regarding the Mediterranean not always are the bodies found, and the horror lies in their absence and the missing presence of all lost. What is also important to keep under consideration is that absence and presence are not binary opposites. They are contexts that can, and indeed do, interweave in complex ways. Brian Rappert (2015) explains how the relationship can be relative—the acknowledged absence of *something* upholds that *something* should have been present in the first place (Rappert 2015, p. 7). Drawing this back to the paper, it is important to stay mindful of whose perspective of presences and absences are being considered. What one, observing from the coast a distant boat in distress, may argue as missing people as being sensorially absent may simultaneously be all-too-present for those on the boats themselves.

### 4.3. Vio-lense and Collective Healing

Throughout this research project, a query has been if anything can be gained from the research findings which could support healing post-conflict. I say post-conflict because even for those refugees and asylum seekers fleeing their home countries for reasons not necessarily related to conflict (such as climate) it is very clear that—for many people—trauma, violence, and conflict is a part of their European journeys (Barker and Zajović 2022).

It is evident that there is considerable collective trauma for tens of thousands of people at the hands of European officials. Whilst it is not the focal point of the research, it is nonetheless important to consider ways in which to enhance their own resilience and survival and support collective healing. Of note in this research was the need to find a mode of healing which did *not* insist on people having the capacity to talk about their experiences, which—for reasons we have already explored—is often not possible. We already know senses are useful in trauma recovery situations (May-Benson 2016; McGreevy and Boland 2020). What was clear in the interviews was that even in the direst situations, such as Calais 'Jungle' (when it remained), this instinctive knowledge that controlling one's senses can foster healing was pertinent. Despite the camp itself being a very clear sensorial excess resulting from violent policies and neglect, there was ample evidence of positive, healing sensorial input too. Our senses can feed into our identities and the residents of the camp knew this, as they shared their meals—rich with tastes and smells they remembered from home. This I would argue is a form of self-administered collective healing and one which does not necessitate talking about traumas. It should be noted however that whilst support could be given on suggested sensory practices, the autonomy of the person is a priority—for some people, some sensory exercises (like breathing or closing one's eyes) could remind them of trauma and be distressing rather than healing (CPTSD Foundation 2019). Even so, by reconnecting with the senses which have been violated as part of the violence, there is hope for collective healing.

### 4.4. Limitations

Whilst this paper serves to provide interesting perspectives on violence and senses, there are limitations in the research design and scope of the study. Firstly, that refugees and asylum seekers were excluded from participating—however much due to ethical concerns—means that their own first-hand account of the violence(s) experience is absent. Future research may wish to consider involving people on the move more directly to compliment this study, now that the COVID-19 restrictions have largely eased and digital communications are no longer the only option.

The scope of the study is also limited. The sample size is not vast, and even with participants understanding situations in various settings, the results cannot be generalised broadly across Europe. They also cannot predict how the picture in Europe may shift, as this

happens often dramatically and quickly. Future research may wish to involve alternative methods such as focus groups or surveys to capture a greater quantity of responses.

Finally, when concerning collective healing, this is only of course pertaining to psychological healing. A future paper would also consider the physical health implications of withstanding such violence and the role senses could play here to enhance collective healing—for example the long-term implications of chronic pain and the highly reported issue of skin conditions (Scarry 1985; Sontag 2003; Barker and Zajović 2022).

**5. Conclusions**

When we consider violence from the perspective of a violation of integrity, the senses can be developed by people on the move—and those supporting them—to reaffirm integrity and healing. Vio-lense establishes a clear connection between the types of violence taking place in Europe and the senses. Some violence is in sensory excess, whilst some is in sensory deprivation—and either can be committed overtly or covertly. Examples of sensorial excess are in hearing the screams of those crossing the Aegean sea, or the neglect leading to overflowing toilets in refugee camps in France. Sensorial excess can also include beatings, and the extensive use of tear gas. Examples of sensorial deprivation can be found in forced undressings (so deprivation of warmth) in sub-freezing temperatures, leading to deaths. Sensorial deprivation is also relevant in the silencing and ostracisation of people on the move. Frequently, violence is multilayered and complex, with excess and deprivation being intertwined. At times, the violence is survived by a clear attempt to control one's own senses. This can be empowering and potentially healing in its form—such as the use of a football uniform kit by the young men in the UK to avoid the critical gaze of the police. However, sadly at other times even this sensorial attempt to survive can have catastrophic consequences, such as the case of the new-born baby on the boat whose cry was too dangerous. By examining violence in this manner, we can see how a violation of the senses in multiple ways becomes part of the complex picture of violence, but could also be incorporated into recovery. By learning and having support in retaining agency over senses whilst on their journeys, there is hope for collective healing.

**Funding:** This research continues to be funded by the Economic and Social Research Council [ESRC UK] for a 1 + 3 studentship as part of South West Doctoral Training Partnership (SWDTP) on the Security, Conflict and Human Rights pathway. The grant number is E84021X.

**Institutional Review Board Statement:** This study was approved by the Ethics Committee of University of Exeter (Reference Number 202021-021) for studies involving humans.

**Informed Consent Statement:** Informed consent was obtained from all subjects involved in the study.

**Data Availability Statement:** In the interests of preserving confidentiality given the sensitive nature of this research, data is not publicly available.

**Acknowledgments:** Thanks are given to the friends and colleagues involved in the 11th International Society for Health and Human Rights (ISHHR) conference of 2022, for inspiring conversations and encouragement. Thanks are also given to my supervision team for their continued support throughout this PhD project.

**Conflicts of Interest:** The author declares no conflict of interest.

**Notes**

[1] Compagnies Républicaines de Sécurité (CRS), a part of the Police Nationale Force and used largely for riot control.

[2] This is to ensure the protection of their own identity and any they may have referred to in their interviews. For this reason, and the high level of confidentiality assured, interview transcripts will not be available in this paper. Not all participants have returned their transcripts yet so only those who have done so are referred to directly in this paper.

[3] To protect participant identity, an exact time is not given.

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
