# Peer review of "Vio-lense: A Model for Understanding How Violence and Senses Relate during Refugee Journeys in Europe, and How This in Turn Can Foster Collective Healing"

_socsci, doi:10.3390/socsci12030131_

Round 1
Reviewer 1 Report
The author's ideas in exploring how violence and senses are related in refugee journeys in Europe are interesting. As part of the doctoral research, it shows a promising start. However, I have the following observation and comments:
1. Author mentioned in the introduction that it is complemented with brief ethnographic reflections, but not many reflections were found in the body of the manuscript. Thus, it is appropriate to claim it as a qualitative paper, as mentioned later.
2. The concepts that were introduced in the beginning, i.e., violence, senses and violence, were not adequately described in relation to the refugee context. At present, these look isolated and lack cohesion.
3. Methodology was inappropriately long and can be shortened the much of the content, especially the first two paragraphs.
4. Author mentioned primary methodology, but there is no secondary methodology. Thus, it does not make much sense to use the word primary.
5. Although the topic of the manuscript is highly relevant and exciting, the result section does not show much comprehensive and rich data to justify the argument that the author came up.
6. Discussion is unnecessarily long. Much of its content looks like a review of the literature. Thus, some of the discussion can be shifted at the beginning of the manuscript. As it does not have a literature review section, shifting the review parts from the discussion can form a literature review section.
7. Discussion has to be short and precise.
8. Author has used several long quotes from literature that can increase the similarity index (i.e., can be detected as plagiarism); thus, paraphrasing is encouraged.
9. Some of the results with quotes are discussed in the discussion, which should be shifted to the result section.
10. Add a separate section on the literature review.
Reviewer 2 Report
SOCSCI-2168206- Peer-review report 1
I suggest the author to make the following changes/corrections:
At Abstract:
1.R13- "Method" should be replaced by "Aim".
2.R14- after the word "Europe", the word "Method" must be added.
3. The abstract should have a maximum of 250 words (the abstract should be shortened by 40 words).
At Introduction:
4.R38. Aim and objectives. At the end of the introduction, at R38- State the aim and specific objectives, including any prespecified hypotheses.
5. Attention to fonts. Fonts from the templates must be used. In R29-R43 you used Times New Roman, in the rest of the paper you used Palatino Linotype.
6.R59-R70. The text must be reformulated. For ethical reasons of the teaching profession, the example must be presented without reference to students.
7.R164. A single statement in the text regarding the doctoral thesis is sufficient. I understood from the beginning that this paper is part of a doctoral research project. I suggest replacing "I am completing a PhD" with another wording.
8.R240- “to volunteer and for my undergraduate fieldwork”. Unnecessary personal statement.
At Materials and methods:
9. It's too long. It must be systematized to be easier to understand.
10. Type of study and Study design- Present the type of study and key elements of study design (R380 should be moved to methodology).
11. Setting- Describe the setting, locations, and relevant dates, including periods of recruitment, exposure, follow-up, and data collection.
12. R317. Participants- Give the eligibility criteria, and the sources and methods of selection of participants. R338-R339 must be moved to the Participants.
13. R331. Study size- Explain how the study size was arrived at.
14. R324. Methodology- Describe any methods used to examine group and interactions.
15. Because it is a scientific work, first-person expression must be avoided (R345-R346 must be reformulated).
16. R350-R354- must be reformulated. No personal details.
17.R322-323- must be moved to the end of the methodology, to R377, at Ethical Considerations.
To the Results:
18. All phrases with bibliography should be moved to discussions. The results must be original, so there is no bibliography.
19. R391. Table 1. The title of the table must be added. The table must be mentioned in the text at R390. “….(Table 1)”.
To Discussions:
20. R539-541 must be reformulated. Without..."naïve." Scientific expression must be used, clear, without epithets and personal impressions that cannot be supported by scientific arguments.
21. R545- Vio-lense (Table 1)………..
22. R606-R607- must be reformulated. Without mentioning the doctoral thesis.
23. R590-R594- must be moved to results. All fragments from different interviews, placed between " ", are results and must be moved to the results section.
24. R612-R614- must be moved to results.
25. Fig.1. Very poor-quality figure. It must be redone, and the numbers must be entered in the figure (If you do not have the authors' consent, it cannot be used, it means plagiarism).
26. R836. Limitations- Discuss limitations of the study.
To the Conclusions:
26. Without bibliographic references to conclusions.
27. The conclusions must refer to the results presented in the paper.
To the Bibliography:
28. References must be written respecting the requirements of the MDPI for books, articles, and web resources.
The STROBE checklist was used for reviewing this paper.
I hope that the suggestions I made will help the author to improve the paper!
Thank you for your work!
Reviewer 3 Report
The submitted manuscript is of enormous interest and relevance. It is well worth paying attention to how the senses influence refugees' experience of violence.
Broadening our views on people in asylum and refugee situations can lead to greater professional engagement and improve our practices.
The following are some issues that the authors may wish to consider in order to improve the work presented:
- The abstract should have the basic information in a structured way: objectives, methodology, results,...
- In the introduction there are hardly any bibliographical references to support the assertions made by the authors. In fact, already the first sentence of the manuscript does not incorporate the necessary support in the literature: It is well established that the senses - our abilitites to see, hear, feel/touch, smell and taste - are
an important tool in anxiety management and trauma recovery. This is something that is repeated throughout the introduction and needs to be improved.
- There are important gaps in the methodology: we do not know the approach used from qualitative research, we know the criteria for inclusion of participants but not for exclusion.
- There is also a lack of a table with basic data on the participants: age, gender, location, experiences, etc. Data that serve to outline the sample of participants.
- The description of the data analysis methodology is very poor and needs to be substantially expanded.
- It is not advisable to include participants' quotes in the discussion.
- A section on limitations is not observed.
Round 2
Reviewer 2 Report
The text has improved. It is clearer and easier to read.
Thanks for the work done!
Reviewer 3 Report
The authors have made significant modifications to their manuscript that meet the requirements of this reviewer.